# Efficacy of the tetravalent protein COVID-19 vaccine, SCTV01E: a phase 3 double-blind, randomized, placebo-controlled trial

Ruizhi Zhang[1,15], Junshi Zhao[2,15], Xiaoping Zhu[3,15], Qinghu Guan[1], Shujun Liu[2], Meihong Li[3], Jianghua Gao[4], Jie Tan[4], Feng Cao[5], Beifang Gan[6], Bo Wu[7], Jin Bai[8], Youquan Liu[9], Gang Xie[10], Chi Liu[11], Wei Zhao[12], Lixin Yan[13], Shuping Xu[13], Gui Qian[13], Dongfang Liu[13], Jian Li[13], Wei Li[13], Xuxin Tian[13], Jinling Wang[13], Shanshan Wang[13], Dongyang Li[13], Jing Li[13], Yuhuan Jiao[13], Xuefeng Li[13], Yuanxin Chen[13], Yang Wang[13], Wenlin Gai[13], Qiang Zhou[13] & Liangzhi Xie [13,14] ✉

Evolution of SARS-CoV-2 variants emphasizes the need for multivalent vaccines capable of simultaneously targeting multiple strains. SCTV01E is a tetravalent COVID-19 vaccine derived from the spike protein of SARS-CoV-2 variants Alpha, Beta, Delta, and Omicron BA.1. In this double-blinded placebo-controlled pivotal efficacy trial (NCT05308576), the primary endpoint was vaccine efficacy (VE) against COVID-19 seven days post-vaccination in individuals without recent infection. Other endpoints included evaluating safety, immunogenicity, and the VE against all SARS-CoV-2 infections in individuals meeting the study criteria. Between December 26, 2022, and January 15, 2023, 9,223 individuals were randomized at a 1:1 ratio to receive SCTV01E or a placebo. SCTV01E showed a VE of 69.4% (95% CI: 50.6, 81.0) 7 days post-vaccination, with 75 cases in the placebo group and 23 in the SCTV01E group for the primary endpoint. VEs were 79.7% (95% CI: 51.0, 91.6) and 82.4% (95% CI: 57.9, 92.6), respectively, for preventing symptomatic infection and all SARS-CoV-2 infections 14 days post-vaccination. SCTV01E elicited a 25.0-fold higher neutralizing antibody response against Omicron BA.5 28 days post-vaccination compared to placebo. Reactogenicity was generally mild and transient, with no reported vaccine-related SAE, adverse events of special interest (AESI), or deaths. The trial aligned with the shift from dominant variants BA.5 and BF.7 to XBB, suggesting SCTV01E as a potential vaccine alternative effective against present and future variants.

The SARS-CoV-2 virus pandemic has resulted in significant morbidity and mortality across the globe. As of Aug 9, 2023, there have been over 769 million confirmed cases of COVID-19 and 6.9 million deaths reported[1]. The ongoing COVID-19 pandemic poses a significant threat to human health and the economy, impacting individuals of all age groups. The vulnerable population, including the elderly and those with pre-existing comorbidities, are particularly susceptible to COVID-19 and face a higher risk of severe disease and mortality, often requiring intensive care[2-7].

Following the emergence of the Omicron variant, it became increasingly evident that multiple variants could prevail concurrently for extended periods[8]. As of July 03, 2023, the prevailing global variant is XBB.1.5, along with other sub-lineages like XBB.1.6, BQ.1, and

BA.2.75[8]. The composition of these variants may vary across different regions/countries at any given time. For example, in the United States, the composition consists of XBB.1.5 (83.15%), XBB (13.60%), BA.2.75 (2.08%), BQ.1 (0.30%), and BA.5 (0.25%)[9], whereas in Australia, the composition includes XBB.1.5 (29.84%), recombinant (27.42%), XBB (23.39%), and BA.2.75 (19.35%)[9]. Concurrently, the incidence of COVID-19-related hospitalizations has shown an upward trend since November 4, 2023. Given the rapid evolution of SARS-CoV-2, developing vaccines with broad-spectrum protection against variants emerging within 6-12 months is an effective strategy to address the evolving pandemic[8,10].

SCTV01E is a tetravalent COVID-19 protein vaccine based on trimeric spike extracellular domain (S-ECD) of SARS-CoV-2 variants Alpha, Beta, Delta, and Omicron BA.1, and adjuvanted with a squalene-based oil-in-water emulsion SCT-VA02B. SCTV01E remained stable at 25 °C for over six months, making it suitable for remote and resource-poor settings. Preclinical studies of SCTV01E have shown broad-spectrum neutralizing potencies against pre-Omicron variants (D614G, Alpha, Beta, and Delta) and newly emerging Omicron subvariants (BA.1, BA.1.1, BA.2, BA.3, and BA.4/5) in naïve BALB/c and C57BL/6 J mice[10]. An immunogenicity and safety phase 3 clinical trial (NCT 05323461) in population previously vaccinated with either inactivated or mRNA COVID-19 vaccine or previously diagnosed with COVID-19 showed that SCTV01E was well-tolerated among 1800 participants and elicited significantly higher levels of neutralizing antibodies against Omicron BA.1 and BA.5 variants than BNT162b2 and BBBIP-CorV on day 28[11,12]. Specially, both BNT162b2 and BBBIP-CorV were developed based on the original SARS-CoV-2 variant. Following these positive outcomes[11,12], on March 22, 2023, SCTV01E received Emergency Use Authorization from the National Health Commission of the People's Republic of China as a booster dose for those who had received primary series COVID-19 vaccination, as well as a primary dose for individuals who have already been infected.

On the basis of the favorable results from the safety and immunogenicity trial, the Phase 3 pivotal efficacy trial was initiated in late December 2022 to assess the effectiveness and safety of SCTV01E in preventing SARS-CoV-2 infection. This trial was conducted during the peak of the COVID-19 pandemic in mainland China, aligning with the transition of the predominant viral variants from BA.5 and BF.7 to XBB[13], leading to rapid case accumulation. As the cutoff date of efficacy follow-up (May 10, 2023), the mean (standard deviation [SD]) follow-up was 127.1 (4.3) days. A sufficient number of pre-defined COVID-19 cases had been gathered for primary endpoint analysis. This report presents the primary analysis results of this pivotal Phase 3 trial. Continued monitoring for long-term safety is ongoing and will extend for 12 months following vaccination.

## Results
### Participants
Between December 26, 2022 and January 15, 2023, a total of 11,010 participants were initially screened for eligibility. Of them, 9223 individuals were randomized at a 1:1 ratio to receive either SCTV01E or placebo (Fig. 1). The Full Analysis Set (FAS) included 9196 participants, with 4595 in the placebo group receiving one dose of normal saline and 4,601 in the SCTV01E group receiving one dose of SCTV01E. Among the FAS population, 6944 (75.5%) participants were between the ages of 18 and 59 years, while 2252 (24.5%) were 60 years and older. 5,274 (57.4%) of participants were male. The demographic characteristics of the participants were well-balanced in the two groups (Table S1). The mean (SD) body mass index (BMI) was 24.1 (3.5) in the placebo group, and 24.2 (3.5) in the SCTV01E group, respectively. Most participants in both groups tested negative for SARS-CoV-2 nucleic acid at baseline, with 3877 (84.4%) in the placebo group and 3961 (86.1%) in the SCTV01E group, while the results were missing for a small number of participants (3 in the placebo group and 6 in the

SCTV01E group). The baseline total anti-spike IgG levels were measured in both the placebo group and the SCTV01E group. In the placebo group, the mean (SD) baseline total anti-spike IgG was 1690.7 (1421.5) BAU/mL, while in the SCTV01E group, it was 1693.7 (1420.3) BAU/mL. A total of 1638 participants (35.6%) in the placebo group and 1650 participants (35.9%) in the SCTV01E group had a baseline total anti-spike IgG level below 338 BAU/mL (The 338 BAU/mL IgG level was selected as an indicator of recent infection.). 3406 (74.1%) participants in the placebo group and 3400 (73.9%) participants in the SCTV01E group had received an inactivated COVID-19 vaccine. The mean (SD) time since last vaccination were 13.2 (2.95) months in the placebo group, 13.1 (2.98) in the vaccine group, respectively. 967 (21.0%) participants in the placebo group and 1017 (22.1%) in the SCTV01E group had a pre-existing comorbidity. The Per-Protocol Efficacy (PPE) set comprised participants who had negative results on both the nucleic acid test and rapid antigen test during screening, and whose baseline levels of anti-spike IgG were less than 338 BAU/mL. At the cutoff date, a total of 1309 participants in the placebo group and 1314 participants in the SCTV01E group were included in the PPE analysis. The demographic characteristics of PPE (Table S2) were also well-balanced in the two groups. All participants received COVID-19 vaccines based on the ancestral strain before enrollment, and their prior vaccination is provided in Table S3.

### Efficacy
The primary endpoint analysis in the PPE population revealed that a total of 98 individuals confirmed symptomatic SARS-CoV-2 infection within 7 days to 4 months post-vaccination by EAC, with most of the cases were collected over a relatively short period. Among these cases, 75 were observed in the placebo group while 23 were confirmed in the SCTV01E group. These findings indicated a vaccine efficacy (VE) of 69.4% (95% CI: 50.6, 81.0), meeting the predefined criteria for success in the protocol (lower 95% CI > 30%) (Fig. 2A). The calculated VE (95% CI) by incorporating the baseline IgG levels as a covariate is 69.7% (51.1, 81.2), indicating minimal numerical differences from the original results (69.4% [50.6, 81.0]) obtained without considering baseline IgG titers as a covariate. VE (95% CI) was 60.7% (43.9, 72.5) for prevention of all SARS-CoV-2 infections (including asymptomatic infection) starting 7 days post-vaccination (Fig. 2B). Measures of VE of 14-day post-vaccination showed VE (95% CI) of 79.7% (51.0, 91.6) in preventing symptomatic SARS-CoV-2 infection (33 cases in the placebo group and 7 in the SCTV01E group) (Figs. 2C), 82.4% (57.9, 92.6) in preventing all SARS-CoV-2 infection (37 in the placebo group and 7 in the SCTV01E group) (Fig. 2D), respectively, indicating an enhanced level of protection at this stage.

The efficacy analysis of SCTV01E in the per-protocol analysis population (PPS) reveals a VE of 60.2% (95% CI, 40.0, 73.6) in preventing symptomatic SARS-CoV-2 infections and 50.9% (95% CI, 33.3, 63.9) in preventing all SARS-CoV-2 infections 7 days after vaccination (Figure S1A and Figure S1B). The VE in preventing symptomatic SARS-CoV-2 infections and all SARS-CoV-2 infections increased to 66.0% (95% CI, 32.6, 82.9) and 66.4% (95% CI, 37.0, 82.1), respectively, 14 days post-vaccination (Figure S1C and Figure S1D).

Subgroup analysis was conducted among the population in the PPE set. The results indicate that VE against symptomatic SARS-CoV-2 infection 7 days post-vaccination was similar in participants aged 18 to 59 years (69.4%, 95% CI: 47.3, 82.3) and those aged 60 years and above (69.2%, 95% CI: 15.7, 88.7) (Fig. 3A). The VE was also similar between the group previously vaccinated with an inactivated vaccine (67.0%, 95% CI: 40.7, 81.6) and the non-inactivated vaccine group (73.4%, 95% CI: 38.9, 88.4). The vaccine demonstrated higher efficacy among participants with pre-existing comorbidities (76.3%, 95% CI: 36.8, 91.1) compared to those without pre-existing comorbidities (66.1%, 95% CI: 41.2, 80.4). Participants who received the study vaccine within 6–12 months of their previous COVID-19 vaccination had a VE (95% CI) of 77.4% (45.2,

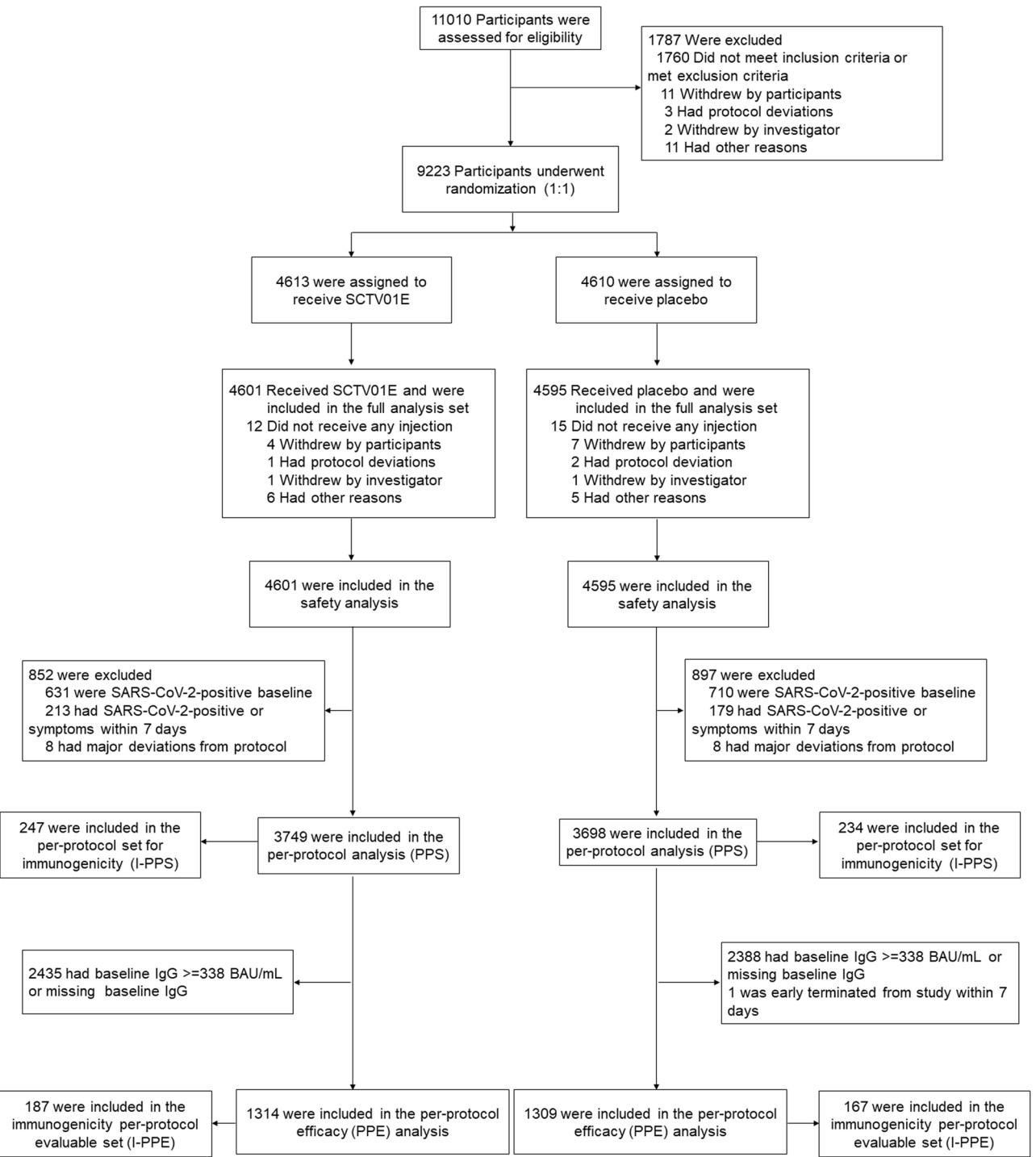

**Fig. 1 | Randomization and analysis population.** 9223 participants were recruited and randomized in this trial. 9,196 participants who received at least one vaccine or placebo were included in the safety set. Per-protocol set had 7447 participants without major deviations. 2623 participants were included in the per-protocol set for efficacy.

90.7), while those with an interval of 12–24 months had a VE of 65.3% (38.6, 80.4).

The VE (95% CI) of SCTV01E against symptomatic SARS-CoV-2 infection 14 days after vaccination was found to be 74.5% (37.4, 89.6) in participants aged 18 to 59 years and 83.6% (−34.9, 99.6) in participants aged 60 years and above (Fig. 3B). SCTV01E showed VE of 100.0% (50.3, 100.0) among participants who were previously vaccinated with a non-inactivated vaccine. Among participants who had previously received an inactivated vaccine, the VE was 73.4% (34.5,

89.2). SCTV01E showed VE of 73.3% (−40.4, 97.3) among participants with pre-existing comorbidities and 83.2% (51.4, 94.2) among participants without pre-existing comorbidities. The VE was 85.1% (49.6, 95.6) in the group who received their last vaccination 12–24 months earlier and 71.2% (−4.9, 92.1) in the group who received their last vaccination 6–12 months earlier.

As of the cutoff date, there was one observed case of moderate COVID-19, which occurred in the placebo group. No severe cases of COVID-19 or deaths related to COVID-19 were reported in this study.

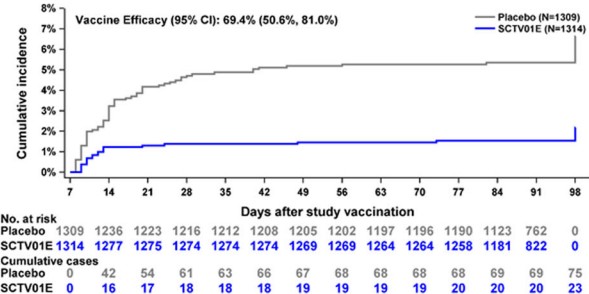

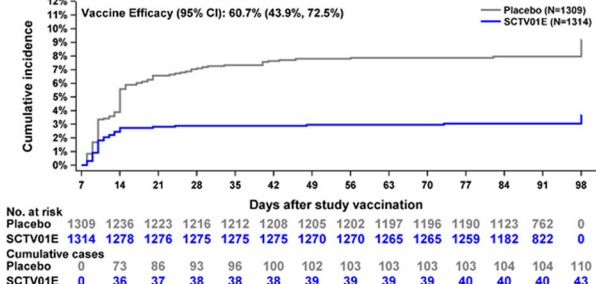

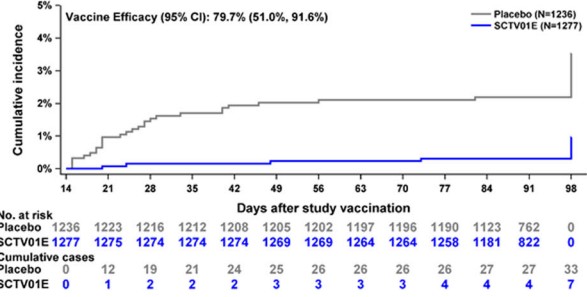

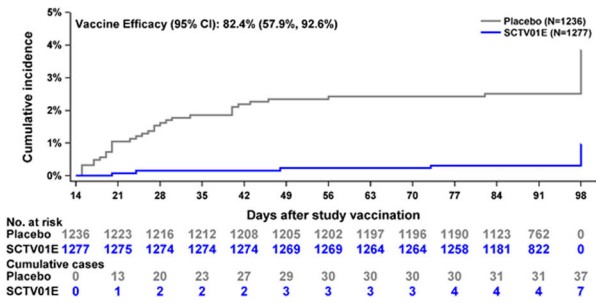

**Fig. 2 | Vaccine efficacy analysis.** Cumulative incidence for COVID-19 cases at 7 and 14 days post-injection of SCTV01E or placebo among population in PPE, were depicted in Panels (**A**) (symptomatic COVID-19 7 days post injection), **B** (all infections 7 days post injection), **C** (symptomatic COVID-19 14 days post injection), and **D** (all infections 14 days post injection).

Compared to the control group, participants who were infected with SARS-CoV-2 in the SCTV01E group experienced a shorter duration (days) of illness (SCTV01E vs. Placebo = 1.0 vs. 3.5, $P = 0.0941$) and fewer COVID-19-related symptoms (SCTV01E vs. Placebo = 1.0 vs. 2.0, $P = 0.0495$) (Table S4).

A post hoc analysis was conducted on the swab samples obtained from the individuals with SARS-CoV-2 infection to assess viral load and conduct viral genotyping specifically related to COVID-19. The viral load test indicated a lower SARS-CoV-2 viral load in individuals who received SCTV01E compared to those who were given the placebo. In the FAS set, the median copies of SARS-CoV-2 in the placebo group were 47.9-fold higher than the SCTV01E group when analyzed by the ORF1AB gene, and 14.5-fold higher when analyzed by N gene (Table S5). The genotyping performed on 29 nasal/nasopharyngeal/throat swab samples revealed a diverse range of SARS-CoV-2 variants/sub-variants (Figure S2). Among the samples, 41.4% were identified as Omicron BA.5.2 sub-lineages (12 cases in control group versus 0 case in the SCTV01E group), 34.5% as Omicron BF.7 sub-lineages (9 cases in control group versus 1 case in the SCTV01E group), and 24.1% as Omicron XBB.1 sub-lineages (6 cases in control group versus 1 case in the SCTV01E group).

## Immunogenicity

The immunogenicity analysis of SCTV01E within the per-protocol set for immunogenicity (I-PPS) revealed the GMT (95% CI) of nAb against Omicron BA.5 was 167 (138, 203), with 18.4-fold change over baseline at 14 days post-vaccination, and 150 (124, 181) with 16.5-fold change over baseline at 28 days post-vaccination (Fig. 4A). The corresponding seroresponse rates (SRR) (95% CI) were 79.8% (74.2%, 84.7%) and 78.6% (73.0%, 83.6%) respectively. Among the participants received SCTV01E in the immunogenicity per-protocol evaluable (I-PPE) set, the geometric mean tier (GMT) (95% CI) of live virus neutralizing antibody (nAb) against Omicron BA.5 were 131 (104, 165) with 30.2-fold change over baseline 14 days post-vaccination, 123 (98, 155) with 28.6-fold

change over baseline 28 days post-vaccination (Fig. 4B). SRR of nAb compared with the pre-injection baseline (95% CI) were 92.4% (87.6%, 95.8%) 14 days post-vaccination, 92.6% (87.8%, 95.9%) 28 days post-vaccination, respectively. SCTV01E elicited a 25.0-fold higher of the nAb responses against Omicron BA.5 28 days post-vaccination comparted to placebo.

## Safety

The safety analysis set consisted of 9196 participants, with 4595 in the placebo group and 4601 in the SCTV01E group. The incidence rates of treatment-related adverse events (TRAEs) were 6.2% and 16.8% in the placebo and SCTV01E groups, respectively. Within seven days after injection, 159 (3.5%) participants in the placebo group and 679 (14.8%) in the SCTV01E group reported at least one solicited local adverse reaction (AR), while 146 (3.2%) and 246 (5.3%) participants in the placebo and SCTV01E groups reported solicited systemic ARs, respectively. Table S6 showed Treatment-Emergent Adverse Events (TEAEs) in participants based on their results of nucleic acid/rapid antigen tests at baseline.

The most frequent (incidence ≥1%) solicited local ARs were injection site pain (SCTV01E vs. placebo, 14.0% vs. 3.0%), injection site pruritus (SCTV01E vs. placebo, 2.6% vs. 0.7%), and injection site swelling (SCTV01E vs. placebo, 1.8% vs. <0.1%) (Fig. 5A). The most frequent solicited systemic ARs were fatigue (SCTV01E vs. placebo, 2.3% vs. 1.0%), headache (SCTV01E vs. placebo, 2.0% vs. 1.1%), myalgia (SCTV01E vs. placebo, 2.0% vs. 0.7%) and pyrexia (SCTV01E vs. placebo, 1.2% vs. 0.9%) (Fig. 5B). Most solicited ARs were Grade 1 or 2, only 6 (0.1%) participants in the SCTV01E group had Grade 3 or above local adverse reactions (ARs), with 4 reporting injection site pain and 2 reporting injection site swelling (Fig. 5A). 15 (0.3%) participants in the SCTV01E group and 14 (0.3%) participants in the placebo group reported Grade 3 or above systemic ARs, including fatigue, headache, myalgia, pyrexia, arthralgia, chills, and nausea (Fig. 5B). All participants recovered before the last safety follow-up. Within 28 days of

## A. Subgroup analysis seven days post injection

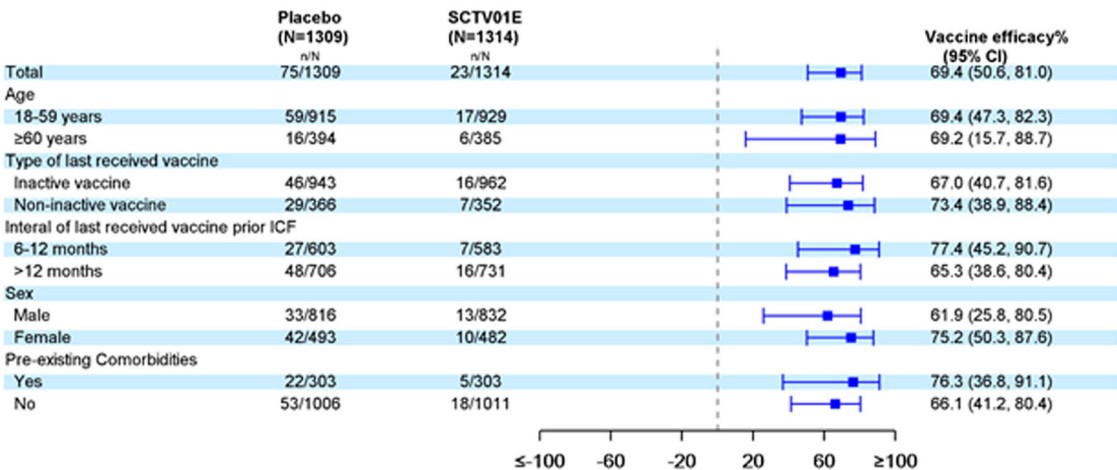

## B. Subgroup analysis fourteen days post injection

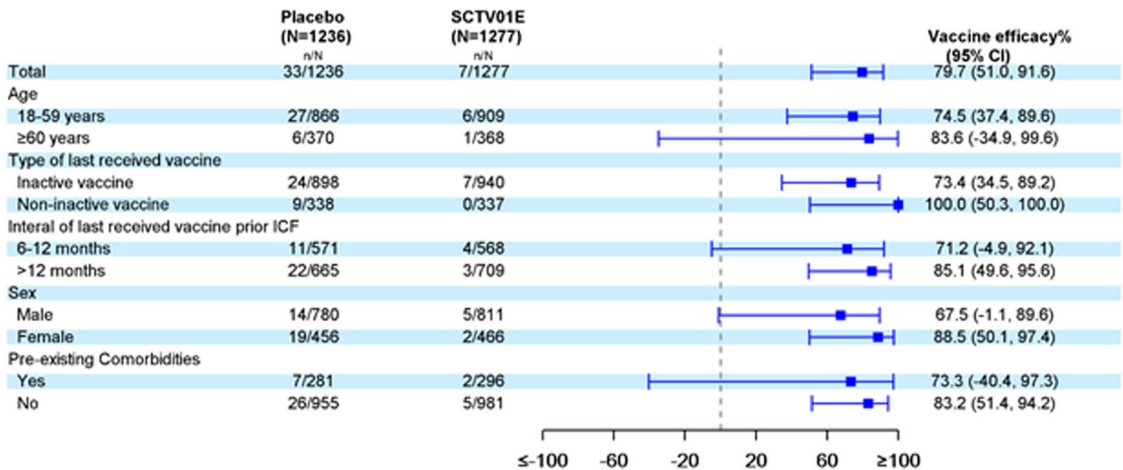

**Fig. 3 | Subgroup analysis of the vaccine efficacy against symptomatic SARS-CoV-2 infection.** Subgroup analysis in Panel (**A**) involved people in the per-protocol set for efficacy who had negative result of nucleic acid test at baseline, anti-spike antibodies <338 BAU/mL, and no SARS-CoV-2 infection within 7 days post-vaccination. Panel (**B**) analyzed a similar subgroup in the per-protocol set for efficacy, including the participants who had no SARS-CoV-2 infection within 14 days post-vaccination. Inactive vaccine, also known as an inactivated vaccine, is a type of vaccine that uses a killed version of the virus. Non-inactive vaccine in this study included protein-based vaccine and adenovirus-vector vaccine. Note: Bars represent the VE, and error bars indicate 95% confidence intervals.

receiving the study vaccine, both the placebo and SCTV01E groups had a similar incidence of unsolicited ARs, with 33 (0.7%) and 47 (1.0%) participants respectively.

Serious adverse events (SAEs) were reported by 0.9% of participants in the SCTV01E group and 0.8% in the placebo group, but none were related to the study vaccine as determined by the investigator. There was one death reported in the placebo group due to a car accident. No adverse events of special interest (AESIs) were observed during the safety follow-up.

In the SCTV01E group, the stratified analysis by age revealed that the occurrence of TRAEs was lower in participants aged 60 years and older compared to those aged 18–59 years. Specifically, among participants aged ≥60 years, the incidence of TRAEs in the SCTV01E group was 9.4% compared to 19.3% in participants aged 18-59 years. Furthermore, the incidences of specific solicited TRAEs in the SCTV01E group were lower among participants aged ≥60 years than among those aged 18-59 years. No Grade 3 or higher solicited local ARs and vaccine-related SAEs were reported in the elderly participants.

The occurrence of TRAEs in participants with pre-existing comorbidities was similar to that observed in the overall safety analysis population, with 6.0% incidence in the placebo group and 16.7% incidence in the SCTV01E group. The incidences of TRAEs in the SCTV01E group were similar among the elderly participants with (10.4%) or without pre-existing comorbidities (9.1%). Similarly, for participants aged 18–59 years, TRAEs among those with or without pre-existing comorbidities were also similar (19.1% vs.19.3%).

## Discussion

This phase 3 clinical trial, which involved 9196 participants, demonstrates the safety and effectiveness of SCTV01E as a heterologous booster dose in preventing SARS-CoV-2 infection. This study was initiated in late December 2022, during the peak of the extensive COVID-19 outbreak in mainland China. According to data reported from December 9, 2022, to January 30, 2023, by the Chinese Center for Disease Control and Prevention on February 1, 2023, the estimated overall infection rate was 87.5% during this period[14]. The protocol

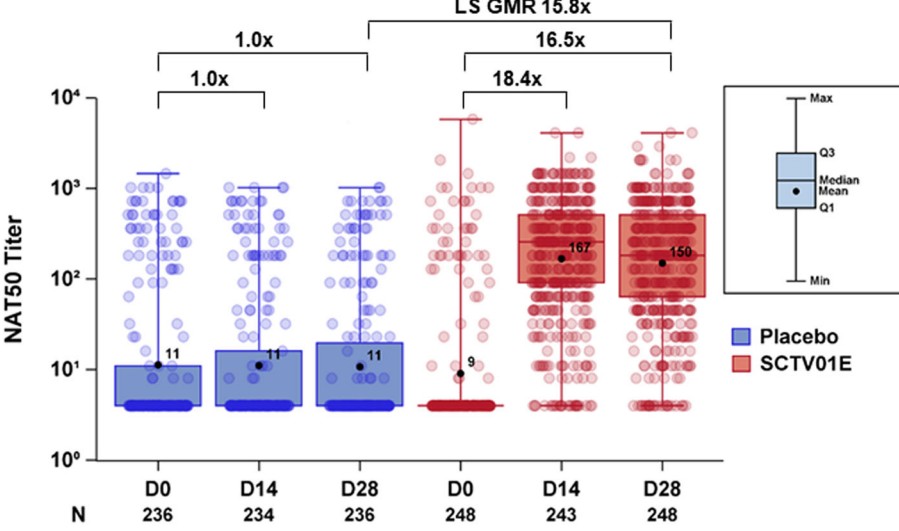

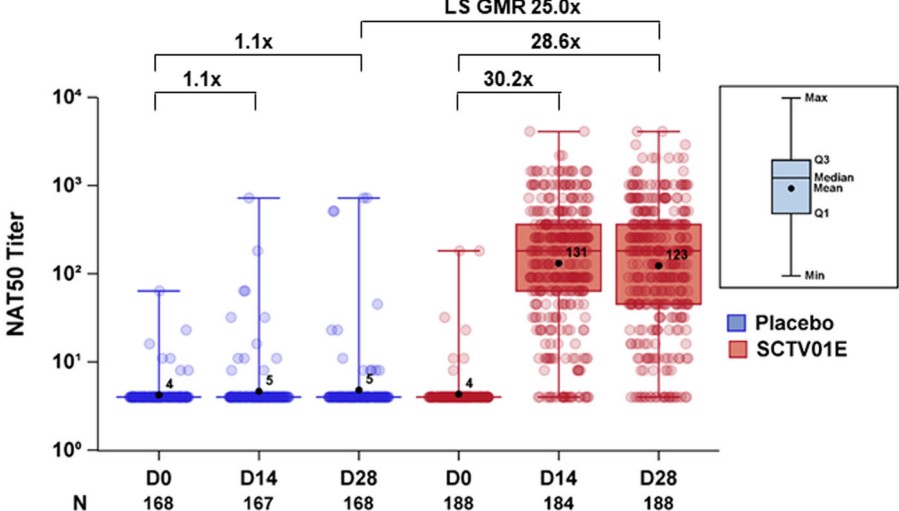

**Fig. 4 | GMTs of live virus nAb against Omicron BA.5.** Panel (**A**) analyzed the geometric mean titers (GMTs) of neutralizing antibodies against Omicron BA.5 at days 14 and 28 post-vaccination using the 50% plaque reduction neutralization test. The study focused on participants in the immunogenicity per-protocol set (I-PPS) who received one dose of study vaccine without significant protocol deviations. Panel (**B**) examined similar data but within the immunogenicity per-protocol evaluable set (I-PPE), considering participants with negative baseline nucleic acid tests and anti-spike antibodies <338 BAU/mL. Note: The box represents the interquartile range (IQR), the bar inside the box represents the median level, and the dot represents the GMT level. The upper and lower whiskers represent the minimum to maximum range of the titer for each group. GMTs and geometric mean fold-rises from baseline (×), and the least square geometric mean ratios (LS GMR) of SCTV01E/Placebo are displayed above the corresponding data points.

incorporated a specific threshold for total anti-spike IgG levels, as recommended by the Center for Drug Evaluation in China. This threshold was used to exclude participants who recently recovered from a SARS-CoV-2 infection from the primary efficacy analysis. The Per-Protocol Efficacy set, comprising participants with IgG levels below 338 BAU/mL, was used to analyze the primary efficacy endpoint. The determination of the cutoff IgG value was based on outcomes of our early phase trials assessing the immunogenicity of protein-based vaccines (NCT05043285, NCT05043311, NCT05323461), alongside continuous monitoring and analysis of dynamic IgG levels among patients during the China outbreak. These findings suggest a substantial association between anti-spike IgG levels and the progression of SARS-CoV-2 infection. In particular, when IgG levels surpass 338 BAU/ml (Which is

10 times the lower limit of detection), there is a high probability of a recent infection.

Currently, several new vaccines were developed and authorized in coping with predominant of Omicron variants/subvariants, including Spikevax bivalent Original/Omicron BA.4-5 and Comirnaty Original/Omicron BA.4-5. The authorization for these bivalent vaccines was based on nonclinical data, as well as safety and immunogenicity data, rather than a pivotal efficacy trial[15,16].

The results showed that, a single dose of SCTV01E provided a VE against symptomatic SARS-CoV-2 infection of 69.4% (95% CI: 50.6, 81.0) in the 7 days post-vaccination. The findings demonstrate success according to the pre-specified criteria in the protocol and surpass the critical criteria of the WHO Target Product Profile for

## A. Solicited Local Adverse Reactions

## B. Solicited Systemic Adverse Reactions

**Fig. 5 | Solicited local and systemic adverse reactions.** Safety analysis was conducted among participants in the full analysis set who had received at least one dose of SCTV01E or Placebo. The incidence and severity of solicited local Panel (**A**) and systemic reactions Panel (**B**) through 7 days after the study vaccination. For each category, adverse reactions were classified as follows: grade 1, mild; grade 2, moderate; and grade ≥3, severe and above.

COVID-19 vaccines ( ~ 50% point estimate and lower 95% CI ≥ 30%)[17,18]. Consistently, SCTV01E elicited a 25.0-fold higher of the nAb responses against live Omicron BA.5 28 days post-vaccination compared to the placebo.

The infected participants in the SCTV01E groups exhibited significantly lower viral load, a shorter duration of illness, and fewer COVID-19-related symptoms compared to participants in the control group. Notably, these positive outcomes were observed during a period when a blend of Omicron BA.5.2, BF.7, and XBB variants/sub-variants were in circulation. Examination of swab samples taken from patients revealed a diverse range of Omicron sub-lineages, including BA.5.2.48/49, DY.1/2/3/4, BF.7.14, XBB.1.16 BF.7.14.1/4/5, XBB.1.9.1, and FL.2. Among the samples, 41.4% were identified as Omicron BA.5.2 sub-lineages, 34.5% as Omicron BF.7 sub-lineages, and 24.1% as Omicron XBB.1 sub-lineages.

The trial assessed efficacy starting from 7 days post-vaccination as the primary endpoint, instead of the typical 14-day period[19-23], due to the nature of the pandemic and based on communication with regulatory authorities. Considering the rapidly progressing SARS-CoV-2 pandemic, it was essential to develop a vaccine that demonstrated efficient efficacy within a short period after vaccination. This decision is also supported by related studies on COVID-19 vaccine booster doses, where both BNT162b2[24] and ZF2001[25] showed favorable efficacy profiles 7 days post-vaccination. Despite the earlier observation window, the trial demonstrated significant VE endpoints, indicating rapid protection from symptomatic infections. The secondary efficacy endpoints evaluated efficacy from 14 days post-vaccination and

showed that VEs for preventing symptomatic infection, and all infection (including asymptomatic SARS-CoV-2 infection) of SARS-CoV-2 were 79.7% (95% CI, 51.0, 91.6), and 82.4% (95% CI, 57.9, 92.6) respectively, indicating an enhanced level of protection at this stage. Considering that there is no statistically significant difference between these two datasets, the numerically high VE against all infections of SARS-CoV-2 compared to that against symptomatic SARS-CoV-2 infection 14 days post-vaccination may largely be due to sampling variability. Likewise, within the per-protocol analysis population consisting of 7447 participants, the VE of SCTV01E vaccine for the prevention all infections of SARS-CoV-2 was 50.9% (95% CI, 33.3, 63.9) 7 days after vaccination, with this efficacy increasing to 66.4% (95% CI, 37.0, 82.1%) 14 days post-vaccination. These findings are in line with other Immunogenicity studies conducted on COVID-19 vaccines. These studies have shown that the level of neutralizing antibody response remained relatively stable until day 4 after vaccination, after which it experienced a rapid surge, reaching its maximum levels around day 14[26].

The subgroup analyses suggest that SCTV01E is effective across different demographic groups, including age, sex, prior COVID-19 vaccine type, interval from the previous dose, and pre-existing comorbidities. Specifically, SCTV01E provides similar efficacy against symptomatic SARS-CoV-2 infection 7 days post-vaccination for adults aged 18-59 and over 60, with VE of 69.4% and 69.2%, respectively. However, due to the limited number of symptomatic SARS-CoV-2 infection cases 14 days post-vaccination among the population with a 6−12 month interval since their last vaccination (11 cases in the placebo

group and 4 cases in the SCTV01E group), the lower limit of the 95% confidence interval for vaccine efficacy (71.2%, 95% CI: −4.9, 92.1) was below zero. Therefore, the changes in vaccine efficacy for populations with shorter (6–12 months) and longer (12–24 months) intervals since vaccination may not accurately reflect the true tendency.

SCTV01E also provides encouraging protection against symptomatic SARS-CoV-2 infection in adults with at least one pre-existing comorbidity, including diabetes, heart disease, respiratory disease, obesity, hypertension, and other conditions, with a VE of 76.3% 7 days post-vaccination and 73.3% 14 days post-vaccination, respectively. This is important as pre-existing comorbidities have been demonstrated to be associated with an increased risk of severe COVID-19 and mortality in adults[5–7].

SCTV01E had a favorable safety profile in this trial and earlier phase studies[27–29], with no vaccination-related safety concerns. The overall incidence of adverse reactions was similar to or lower than that of most approved COVID-19 vaccines[19,30,31], and the majority were mild-to-moderate and transient. Grade 3 or above adverse reactions were rare and occurred similarly between the placebo and SCTV01E groups. No vaccination-related AESI and death were reported. Additionally, in the SCTV01E group, the incidence of adverse reactions was lower among participants aged ≥60 years, and the incidences of TRAEs were similar among participants with (10.4%) or without pre-existing comorbidities (9.1%). These findings support the safety of protein-based vaccines, which have been extensively used to prevent various diseases such as hepatitis B and C, influenza, pertussis, and human papillomavirus[32].

The trial has several limitations. Firstly, the study was conducted during the COVID-19 outbreak, and a substantial number of the participants showed IgG levels exceeding 338 BAU/ml, indicating a high likelihood of recent infection. These recent infections could potentially influence the safety evaluation. Secondly, due to significant COVID-19 outbreaks in the trial region, the primary endpoint cases were collected over a relatively short period. Long-term follow-up data will be gathered for at least 6-12 months to address this. Thirdly, due to the low number of moderate and severe COVID-19 cases associated with the Omicron variant, it was not possible to fully assess the VE of SCTV01E against these severities by the data cutoff date. Fourthly, comparing our vaccine to saline rather than an inactivated or mRNA vaccine may limit the study's impact. The decision for a placebo-controlled design was influenced by the challenges of licensed COVID-19 vaccines in China, given the limited available evidence supporting their efficacy against the new variants during the study period.

In summary, the phase 3 trial evaluated SCTV01E, a tetravalent protein-based COVID-19 vaccine, as a heterologous booster dose during the circulation of Omicron variants/subvariants. SCTV01E demonstrated sufficient efficacy and safety, suggesting its potential as a vaccine alternative for preventing symptomatic SARS-CoV-2 infection.

## Methods
### Study design
This phase 3 study was a randomized, double-blind, placebo-controlled evaluation of the efficacy and safety of SCTV01E, a tetravalent SARS-CoV-2 trimeric spike protein vaccine, in adults aged 18 years and above. Participants were recruited from 9 study sites located across three provinces in China, including Sichuan, Guizhou, and Hunan. The protocol of this study, the written informed consent form, and other information related to participants were approved by the clinical research ethics board of the Sichuan/ Guizhou/Human Provincial Center for Disease Control and Prevention (China). This trial followed the Declaration of Helsinki, Good Clinical Practice (GCP) requirements, and related regulations issued by authorities.

### Participants
Eligible participants were adults aged 18 years or above who had previously received the primary series of COVID-19 vaccine or booster vaccination, with a 6 to 24-month vaccination interval between the last dose COVID-19 vaccine and the study vaccination. Participants were excluded if they had a fever (temperature ≥37.3°C) within three days before the study vaccination, a history of SARS-CoV-2 infection within 6 months, or a positive result for nasal/nasopharyngeal/throat swab nucleic acid test or rapid antigen test during screening. Full details related to the inclusion and exclusion criteria are provided in the trial protocol.

### Randomization and masking
This study was conducted as a double-blind trial. Everyone involved, including participants, investigators, clinical research associates, data analysts, and laboratory staff, remained unaware of group assignments. Non-blind teams handled tasks such as vaccine reception, management, allocation, injection, packaging recovery, and overall drug management. They were strictly prohibited from participating in any study-related assessments or contacting participants for data collection after vaccine administration. Interactive Network Response System (IWRS) was used to randomize the eligible participants prior to study vaccination, they were stratified by age (aged 18-59 years, ≥60 years), and the type of COVID-19 vaccine last received (inactivated vaccine, non-inactivated vaccine). The randomization codes were generated via block randomization using SAS software (Version 9.4).

### Procedures
Eligible adults were randomly assigned in a ratio of 1:1 to receive one dose of SCTV01E or placebo. Both placebo and SCTV01E were given as intramuscular injections in the deltoid muscle on the outside of the upper arm. After receiving the vaccination, all participants were observed at the site for at least 30 minutes after the study intervention. Both active monitoring and spontaneous reporting were used. Solicited AEs within 7 days, and unsolicited AEs within 28 days after the study vaccination were collected through vaccination record cards (VRCs). Serious adverse events (SAEs) and adverse events of special interest (AESIs) were collected by investigators via phone calls, short messages, e-mails, visits on-site, or other contact methods for up to 365 days. SAEs and AESIs are still being monitored and collected after this efficacy analysis.

After the study vaccination, participants were contacted weekly via phone calls, text messages, emails, or other means of communication to inquire about any signs or symptoms related to COVID-19. Additionally, participants were encouraged to report any COVID-19-related symptoms they experienced spontaneously at any time during the study period.

Two samples of nasal/nasopharyngeal/throat swabs were collected from participants who showed any signs or symptoms of COVID-19. One sample for the antigen rapid test and/or SARS-CoV-2 reverse-transcriptase–polymerase-chain-reaction (RT-PCR) test, the other for virus sequencing. If the results of the antigen rapid test and /or RT-PCR were positive, the virus was isolated from the nasal/naso-pharyngeal/oropharyngeal swab, and viral sequencing was used to identify the major SARS-CoV-2 variants. Primary endpoint cases were defined as occurring in participants who had at least one of the following symptoms: fever, chills, sore throat, general weakness/fatigue, myalgia, headache, anorexia/nausea/vomit, diarrhea, olfactory disorder/taste disorder, cough/expectoration, shortness of breath, difficulty breathing, clinical or radiographic evidence of pneumonia, and at least one nasopharyngeal swab, nasal swab, or saliva sample that was positive for SARS-CoV-2 by antigen test and/or RT-PCR test. The trial protocol provides details of the diagnostic criteria and definitions for symptomatic and infection with multiple symptoms COVID-19 cases.

An Endpoint Adjudication Committee (EAC) was established to conduct independent evaluation and judgment for each case (Figure S3).

Blood samples were collected from participants in the immunogenicity subgroup on 0 (pre-injection), 14, and 28 days post-vaccination. Additional blood samples were collected from 200 participants in the immunogenicity subgroup on 7, 90, 180, and 365 days post-vaccination for geometric mean titers (GMTs) of live virus neutralizing antibodies against Omicron BA.5 variant using 50% plaque reduction neutralization test (PRNT50).

## Outcomes

The primary endpoint of this study was cases of the first occurrence of symptomatic infection of SARS-CoV-2 of any severity starting 7 days post-vaccination. The second efficacy endpoints included: cases of the first occurrence of all infection (including asymptomatic infection), asymptomatic infection, infection with multiple symptoms, moderate and above, severe and above COVID-19 and death due to COVID-19, respectively, starting 7 days post-vaccination; cases of the first occurrence of all infection (including asymptomatic infection), asymptomatic infection, symptomatic infection, and infection with multiple symptoms, respectively, starting 14 days (≥15 days) post-vaccination; cases of the first occurrence of symptomatic infection, infection with multiple symptoms, moderate and above, severe and above COVID-19 and death due to COVID-19, respectively, caused by SARS-CoV-2 variants and subvariants starting 14 days (≥15 days) post-vaccination.

The immunogenicity endpoints in the study included: GMT of neutralizing antibody (nAb) against SARS-CoV-2 variants or subvariants on 7, 14, 28, 90, 180, and 365 days post-vaccination; the seresponse rates (SRRs) of nAb (change from <lower limit of quantification (LLOQ) to ≥4 ×LLOQ, or at least a fourfold rise if baseline ≥LLOQ) compared with the pre-injection baseline (95%CI) against SARS-CoV-2 variants or subvariants on 7, 14, 28, 90, 180 and 365 days post-vaccination.

The safety endpoints in the study included: the incidence and severity of solicited AEs of SCTV01E from Day 0 to Day 7, incidence and severity of unsolicited AEs from Day 0 to Day 28; incidence and severity of serious AEs and AESIs from Day 0 to Day 365. The severity of AEs was graded according to the criteria of the Toxicity Grading Scale for Healthy Adult and Adolescent Volunteers Enrolled in Preventive Vaccine Clinical Trials[33].

In the post hoc analysis, the viral load in a patient's sample was quantified using cycle threshold (CT) obtained from the RT-PCR test. The CT value indicates the number of amplification cycles needed for the SARS-CoV-2 RNA sequence to become detectable. To estimate the number of viral particles present in one milliliter of the sample being tested, the CT value was converted into log10 copies/mL using the formula 13.2 − CT* 0.30769[34–36].

## Statistical analysis

For the primary endpoint analysis, the efficacy will be demonstrated if the null hypothesis is rejected when the lower limit of the 2-sided 95% CI of the VE is greater than 30%. A total of 59 symptomatic SARS-CoV-2 infection cases will provide approximately 90% power to detect a VE of 70% at one-sided type one error 0.025. The study is case-driven, approximately 10,000 participants were enrolled in the study to cumulate the target number of symptomatic SARS-CoV-2 infection cases.

In this analysis for efficacy, we present efficacy data at the point of confirmation of 98 symptomatic SARS-CoV-2 infection cases as defined in the protocol. The primary endpoint analysis for efficacy was conducted on the per-protocol efficacy set (PPE), which includes individuals with total anti-spike antibodies <338 BAU/mL at baseline, a negative result on the baseline nasal/nasopharyngeal/throat swab nucleic acid test, and no SARS-CoV-2 infection within 7 days after receiving the study vaccination. The analysis of the other efficacy endpoints was based on the population in PPE and per-protocol set.

The per-protocol set included all the participants who had received one dose of the study vaccine and had no significant protocol deviations. Participants who experienced early infection within 7 days after receiving the dose were excluded from the primary analysis population. For participants who withdrew early, their data were censored at the last assessment date before discontinuation.

The efficacy, defined as a percentage reduction in the hazard ratio, was evaluated using the stratified Cox proportional hazards regression model. The stratification factor included baseline age (18-59 years or ≥60 years), and the type of the last dose of COVID-19 vaccine received (inactivated or non-inactivated). The selection of participant age and the type of COVID-19 vaccine received as covariates in this analysis was driven by their potential impact on the assessment of vaccine effectiveness. Age is critical in assessing COVID-19 severity and outcomes, particularly for individuals aged 60 or above who face higher risks of severe illness and death due to weakened immune function and comorbiditie[37–41]. Previous vaccination can impact the assessment of new vaccines through various mechanisms, including immune system priming, memory immune responses from prior vaccinations, cross-reactivity between components of previous vaccines and antigens in the new vaccine, and competition for immune resources. In China, a large portion of the population has been fully vaccinated with inactivated COVID-19 vaccines[42]. Therefore, previous vaccination status was categorized based on whether individuals received inactivated or non-inactivated vaccines.

The immunogenicity analysis was conducted on the per-protocol set for immunogenicity (I-PPS) and the immunogenicity per-protocol evaluable set (I-PPE), comprising participants from the initial randomized 5% population who received the study vaccine and had no significant protocol deviations. Participants in the I-PPE set had baseline total anti-spike antibodies <338 BAU/mL. Participants who had evidence of SARS-CoV-2 infection and post-infection immunogenicity data were excluded from the analysis, and pre-infection immunogenicity data were still included in the analysis.

To address the multiplicity for multiple comparisons of secondary endpoints, a fixed-sequence hierarchical strategy was employed to control the Type I error rate at one-sided 0.025 (Figure S4) (more details about the endpoints and corresponding sequence, refer to the Multiplicity section of the protocol).

Safety evaluation included all participants who had received one dose of the study vaccine. Participants who received the study vaccine without randomization were excluded from the safety set. Unsolicited adverse events were coded with the Medical Dictionary for Regulatory Activities (MedDRA), version 25.1.

## Reporting summary

Further information on research design is available in the Nature Portfolio Reporting Summary linked to this article.

# Data availability

Data associated with this study are provided in the paper or supplementary materials. Source data for Figs. 4 and 5 and the protocol and statistical analysis plan are provided as Supplementary information. As the trial is still ongoing, anonymized participant data will be made available when the trials are complete, upon requests directed to the corresponding author. Proposals will be reviewed and approved by the sponsor, investigator, and collaborators based on the scientific merit of the request. After approval of a proposal, data can be shared through a secure online platform after signing a data access agreement. Source data are provided in this paper.

# Code availability

All code used to produce the results can be accessed by sending a scientifically sound proposal to LX@sinocelltech.com. Shared code will be made available with the associated raw data.

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

## Acknowledgements

This study was sponsored by Sinocelltech Ltd. and funded by the Beijing Science and Technology Planning Project [Z231100004823003] and the National Key Research and Development Program of China [2022YFC0870600]. The sponsor contributed to trial design, data analyses, and data interpretation. We thank Dr. Jun Han, an employee of State Key Laboratory of Infectious, for his contribution in providing consultation and advice on the exploration of the immunogenicity analysis.

## Author contributions

L.X. contributed to the study conception, design, supervision, and overall management. R.Z., J.Z., and X.Z., the leading site principal investigators, contributed to the study design, supervision, and coordination. Q.G., S.L., M.L., J.G., J.T., F.C., B.G., B.W., J.B., Y.L., G.X., C.L., and W.Z. contributed to the implementation of the study and data collection. D.L., Jing.L., Y.J., and X.L. were responsible for biological sample handling, laboratory testing, and assay development. L.Y., S.X., D.L., Jian.L., and Y.C. contributed to the medical monitoring, data verification, and analysis, results presentation, and interpretation. W.L., X.T., S.W., and J.W. were responsible for the clinical operation and coordination. Y.W., W.G. and Q.Z. contributed to study design, clinical operation, and coordination. G.Q. contributed to writing the manuscripts.

## Competing interests

L.Y., S.X., G. Q., D. L., Jian.L. W.L., X.T., J. W., S.W., D.L., Jing.L., Y.J., X.L., Y.C., Y.W., W.G., and Q.Z. are employees of Sinocelltech Ltd., and X.L., Y.W., W. G., and L.X. have potential stock option interests in the company. R. Z., J.Z., X. Z., Q.G., S.L., M.L., J.G., T.J., F. C., B.G., B.W., J.B., Y.L., G.X., C.L., and W.Z. are investigators and assay technologists involved in this study, and they have no conflicts of interest to declare.

## Additional information

¹Guizhou Center for Disease Control and Prevention, Guiyang, China. ²Hunan Provincial Center for Disease Control and Prevention, Changsha, China. ³Sichuan Provincial Center for Disease Control and Prevention, Chengdu, China. ⁴Songtao Miao Autonomous County Center for Disease Control and Prevention, Tongren, China. ⁵Xiangtan City Center for Disease Control and Prevention, Xiangtan, China. ⁶Shimen County Center for Disease Control and Prevention, Changde, China. ⁷Dejiang County Center for Disease Control and Prevention, Dejiang, China. ⁸Sinan County Center for Disease Control and Prevention (County CDC), Tongren, China. ⁹Mianyang City Center for Disease Control and Prevention, Mianyang, China. ¹⁰Mianyang Youxian District Center for Disease Control and Prevention, Mianyang, China. ¹¹Disease Prevention and Control Center of Yuping Dong Autonomous County, Yuping Dong Autonomous County, Tongren, China. ¹²Nanbu County disease control and prevention center, Nanchong, China. ¹³Beijing Engineering Research Center of Protein and Antibody, Sinocelltech Ltd, Beijing, China. ¹⁴Cell Culture Engineering Center, Chinese Academy of Medical Sciences & Peking Union Medical College, Beijing, China. ¹⁵These authors contributed equally: Ruizhi Zhang, Junshi Zhao, Xiaoping Zhu. ✉e-mail: LX@sinocelltech.com

