## [Peer Review File · Nature Communications]

Efficacy of the Tetravalent Protein COVID-19 Vaccine, SCTV01E: a Phase 3 Double-blind, Randomised, Placebo-controlled TrialEditorial Note: This manuscript has been previously reviewed at another journal that is not operating a transparent peer review scheme. This document only contains reviewer comments and rebuttal letters for versions considered at Nature Communications.

Reviewers' Comments:

Reviewer #1:

Remarks to the Author:

This is an interesting phase 3 placebo controlled efficacy trial of SCTV01E and novel tetravalent protein subunit COVID19 vaccine incorporating the spike protein from 4 SARS-CoV-2 Variants. To my knowledge this is the only such tetravalent vaccine candidate that's been tested in a phase 3 efficacy trial which ensures the novelty of these findings. Though the authors have previously published results from a similar trial conducted during the BA.1 waves the timing of this particular study is during the emergence of more recent SARs-CoV-2 omicron subvariants. It's also a placebo controlled trial which is very challenging at the point and the authors take into account history of prior or recent infection by restricting their PPE analysis to those with low baseline antibody levels. The research design is sound, the results well presented and the conclusions are appropriate. I have a few suggestions for the authors which might strengthen the presentation of the work and bring some clarity for the reader.

Abstract:

Please specify in line 49 that the primary endpoint was efficacy 7 days after vaccination.

It's difficult to understand why efficacy against all infection would be higher than efficacy against symptomatic infection. This might need further discussion.

Introduction:

Line 79-83: This sentence is confusing. Please revise.

Line 95-96: Please specify that these are all first generation vaccines and therefore may not boost as well against BA.1.

Results:

Line 124: I assume what is meant is SARS-CoV-2 nucleic acid.

Line 144: I don't think you want this sentence. Just reference Table S2.

I think you put somewhere in the results the justification to look at the primary endpoint only in those with baseline IgG levels below 338. It's somewhat discussed in the discussion but is very confusing to the reader at this point. It should be mentioned that this is being used as a surrogate for prior infection.

It seems that most of the infections were between days 7-14. This should be pointed out in the results and discussed in the discussion.

The authors never discuss the curious finding that for the 7 day analysis longer time since last vaccination had lower efficacy but this was not true for the 14 day analysis. It was the reverse. This

seems contradictory and non-intuitive and should be mentioned in the discussion.

How were the cases chosen for sequencing and viral load quantification?

Discussion:

The US studies mentioned in lines 299-305 did not exclude patients with prior infection. Therefore the comparison might not be appropriate. Rather it should be compared to the per protocol analysis set (lines 331-335).

Please expand on the justification for the non-standard 7 day post-vaccination primary endpoint when antibody titers are known to peak at Days 14-28.

Figures and Tables:

Figure 4: Solicited Adverse Events - would consider change the y axis scale. It's difficult to see the findings.

Reviewer #2:

Remarks to the Author:

In this study entitled "Efficacy of the Tetravalent Protein COVID-19 Vaccine, SCTV01E: a Phase 3 Double-blind, Randomized, Placebo-controlled Trial," the authors report the efficacy results of the Phase III COVID-19 vaccine trial.

In this study, the authors compare vaccine efficacy of SCTV01E, a tetravalent subunit vaccine containing trimeric spike protein for Alpha, Beta, Delta, and Omicron BA.1 variants adjuvanted with a squalene-based oil-in-water emulsion to placebo in an immune population (vaccinated or previously infected). These studies follow up the results of a phase III immunogenicity trial ([https://www.thelancet.com/journals/eclinm/article/PIIS2589-5370\(23\)00372-3/fulltext](https://www.thelancet.com/journals/eclinm/article/PIIS2589-5370(23)00372-3/fulltext)), which showed enhance neutralization titers relative to inactivated vaccine or bivalent vaccine, and another study comparing SCTV01E to monovalent pfizer vaccine showing improved omicron vaccine induced neutralization titers. The study is well performed and that the manuscript overall is well written.

The author have addressed most of my issues from previous iteration of this manuscript. The major issue below that I have with the study cannot be addressed by the authors, however they have added a sentence to the discussion of why they used a saline control, and provided some al be it flawed) comparison to other studies.

Major Issues

While this study has already been completed and was reviewed by an IRB, I am not sure the ethics of a December 2022 placebo-controlled trial for a COVID-19 booster vaccine. By Dec 2022, it was clear that vaccines prevent infections and disease caused by SARS-COV-2. Whether ethical or not, by comparing to saline, rather than an inactivated vaccine or an mRNA vaccine, even without and updated variant, limit the utility of the results. The real-world decision is not whether to vaccinate, it is what to vaccinate with. Thus, the results of this study have limited impact, as there was no comparator vaccine.

Minor issues

1)The authors should rewrite the sentence Lines 143-145:

Current phrasing

"All participants received COVID-19vaccines based on the ancestral strain before enrollment. As required by the reviewers, the information regarding the prior vaccination is provided in Table S2."

Suggested New phrasing

"All participants received COVID-19 vaccines based on the ancestral strain before enrollment, and their prior vaccination history is provided in Table S2."

2)The vaccination history in Table S2 should sub-grouped based on placebo or treatment group

3) In the discussion the author reference the CDC report of vaccine effectiveness (ref 17) however the data is presented as a comparison to the current study. This should be changed as the 2 studies should not be compared head to head for several reasons, including 1) populations are quite different in terms of vaccine history in USA and China, 2) Assess vaccine efficacy at 2-3 months (ref 17) vs 7 or 14 days in the current study.

Reviewer #3:

None

Reviewer #4:

Remarks to the Author:

The authors have incorporated all the suggested corrections from the earlier version and I have no further comments.

REVIEWERS' COMMENTS

Reviewer #1 (Remarks to the Author):

This is an interesting phase 3 placebo controlled efficacy trial of SCTV01E and novel tetravalent protein subunit COVID19 vaccine incorporating the spike protein from 4 SARS-CoV-2 Variants. To my knowledge this is the only such tetravalent vaccine candidate that's been tested in a phase 3 efficacy trial which ensures the novelty of these findings. Though the authors have previously published results from a similar trial conducted during the BA.1 waves the timing of this particular study is during the emergence of more recent SARs-CoV-2 omicron subvariants. It's also a placebo controlled trial which is very challenging at the point and the authors take into account history of prior or recent infection by restricting their PPE analysis to those with low baseline antibody levels. The research design is sound, the results well presented and the conclusions are appropriate. I have a few suggestions for the authors which might strengthen the presentation of the work and bring some clarity for the reader.

Abstract:

Q1. Please specify in line 49 that the primary endpoint was efficacy 7 days after vaccination.

Response: As per your suggestion, line 49 was revised as: "SCTV01E showed a VE of 69.4% (95% CI: 50.6, 81.0) 7 days post-vaccination, with 75 cases in the placebo group and 23 in the SCTV01E group for the primary endpoint."

Q2. It's difficult to understand why efficacy against all infection would be higher than efficacy against symptomatic infection. This might need further discussion.

Response: Thank you for bringing this concern to us. In this study, the vaccine efficacy of SCTV01E against symptomatic SARS-CoV-2 infection 14 days post-vaccination was numerically lower than that for all infections of SARS-CoV-2 (79.7% [95%CI: 51.0, 91.6] vs 82.4% [95% CI: 57.9, 92.6]) among population in the PPE set. Since there is no statistical difference between these two datasets, this difference may be due to sampling variability.

As per your suggestion, we added the explanation into the revised manuscript as: "*Considering that there is no statistically significant difference between these two datasets, the numerically high VE against all infections of SARS-CoV-2 compared to that against symptomatic SARS-CoV-2 infection 14 days post-vaccination may largely be due to sampling variability.*"

Introduction:

Q3. Line 79-83: This sentence is confusing. Please revise.

Response: As per your suggestion, the sentence has been revised and simplified as: "*Given the rapid evolution of SARS-CoV-2, developing vaccines with broad-spectrum protection against variants emerging within 6-12 months is an effective strategy to address the evolving pandemic*".

Q4. Line 95-96: Please specify that these are all first generation vaccines and therefore may not boost as well against BA.1.

Response: As per your suggestion, we have added the information after line 95-96, as: *“Specially, both BNT162b2 and BBBIP-CorV were developed based on the original SARS-CoV-2 variant.”*

Results:

Q5. Line 124: I assume what is meant is SARS-CoV-2 nucleic acid.

Response: Thank you for bringing this concern for us, we have revised it as: *“Most participants in both groups tested negative for SARS-CoV-2 nucleic acid at baseline”*.

Q6. Line 144: I don't think you want this sentence. Just reference Table S2.

Response: Thanks for your reminder, the sentence has been revised as: *“All participants received COVID-19 vaccines based on the ancestral strain before enrollment, and their prior vaccination is provided in Table S3”*.

Q7. I think you put somewhere in the results the justification to look at the primary endpoint only in those with baseline IgG levels below 338. It's somewhat discussed in the discussion but is very confusing to the reader at this point. It should be mention that this is being used a surrogate for prior infection.

Response: Thank you for your feedback. As per your suggestion, the justification for focusing on the primary endpoint only in those with baseline IgG levels below 338 *BAU/mL* has been shown in the results section. as: *“The 338 BAU/mL IgG level was selected as an indicator of recent infection.”*

Q8. It seems that most of the infections were between days 7-14. This should be pointed out in the results and discussed in the discussion.

Response: As per your suggestion, we added this information in the Result as: *“The primary endpoint analysis in the PPE population revealed that a total of 98 individuals confirmed symptomatic SARS-CoV-2 infection within 7 days to 4 months post-vaccination by EAC, with most of the cases were collected over a relatively short period.”*, and the limitation section of Discussion as *“Due to significant COVID-19 outbreaks in the trial region, the primary endpoint cases were collected over a relatively short period. Long-term follow-up data will be gathered for at least 6-12 months to address this.”*

Q9. The authors never discuss the curious finding that for the 7 day analysis longer time since last vaccination had lower efficacy but this was not true for the 14 day analysis. It was the reverse. This seems contradictory and non-intuitive and should be mentioned in the discussion.

Response: Thank you for pointing out the finding regarding the differences in vaccine efficacy related to the time since the last vaccination between the 7-day and 14-day analyses. For the 14-day analysis, the number of cases in the population with a 6-12 month interval since the last vaccination was limited (11 cases in the placebo group and 4 cases in the SCTV01E group). Consequently, the lower limit of the 95% confidence interval for vaccine efficacy against symptomatic SARS-CoV-2 infection (71.2%, 95% CI: -4.9, 92.1) was below zero, indicating that the observed efficacy may not reflect the true trend.

As per your suggestion, we have revised the discussion to include this point: *“Due to the limited number of symptomatic SARS-CoV-2 infection cases 14 days post-vaccination among the population with a 6-12 month interval since their last vaccination (11 cases in the placebo group and 4 cases in the SCTV01E group), the lower limit of the 95% confidence interval for vaccine efficacy (71.2%, 95% CI: -4.9, 92.1) was below zero. Therefore, the changes in vaccine efficacy for populations with shorter (6-12 months) and longer (12-24 months) intervals since*

vaccination may not accurately reflect the true tendency.”

Q10. How were the cases chosen for sequencing and viral load quantification?

Response: Participants who showed any signs or symptoms of COVID-19 were sampled for 2 samples of nasal/nasopharyngeal/throat swabs. One sample for the antigen rapid test and/or SARS-CoV-2 reverse-transcriptase–polymerase-chain-reaction (RT-PCR) test, the other for virus sequencing. If the results of the antigen rapid test and /or RT-PCR were positive, the virus was isolated from the nasal/nasopharyngeal/oropharyngeal swab and viral sequencing was used to identify the major SARS-CoV-2 variants.

This information has been described in the Section of Procedures, as: *“Two samples of nasal/nasopharyngeal/throat swabs were collected from participants who showed any signs or symptoms of COVID-19. One sample for the antigen rapid test and/or SARS-CoV-2 reverse-transcriptase–polymerase-chain-reaction (RT-PCR) test, the other for virus sequencing. If the results of the antigen rapid test and /or RT-PCR were positive, the virus was isolated from the nasal/nasopharyngeal/oropharyngeal swab and viral sequencing was used to identify the major SARS-CoV-2 variants.”*

Discussion:

Q11. The US studies mentioned in lines 299-305 did not exclude patients with prior infection. Therefore, the comparison might not be appropriate. Rather it should be compared to the per protocol analysis set (lines 331-335).

Response: Thank you for bringing this concern for us. Considering the differences in the populations of the two studies in terms of vaccination history between the USA and China, and the impracticality of comparing vaccine efficacy head-to-head from two different studies, we have deleted this paragraph in the revised manuscript..

Q12. Please expand on the justification for the non-standard 7 day post-vaccination primary endpoint when antibody titers are known to peak at Days 14-28.

Response: Thank you for your inquiry. This study was initiated in late December 2022, during the peak of a significant COVID-19 outbreak in mainland China. Given the rapid spread of SARS-CoV-2, it was crucial to develop a vaccine that could demonstrate efficient efficacy within a short period post-vaccination. Consequently, based on communications with regulatory authorities and the urgent nature of the pandemic, we assessed efficacy starting from 7 days post-vaccination as the primary endpoint, instead of the typical 14-day period, even though neutralizing antibody titers typically peak at Days 14-28. This decision is also supported by related studies on COVID-19 vaccine booster doses, such as BNT162b2 and ZF2001, which have shown favorable efficacy profiles 7 days post-vaccination. As per your suggestion, the manuscript has been revised to include the following justification: *"The trial assessed efficacy starting from 7 days post-vaccination as the primary endpoint, instead of the typical 14-day period^{19, 20, 21, 22, 23}, due to the urgent nature of the pandemic and based on communication with regulatory authorities. Considering the rapidly progressing SARS-CoV-2 pandemic, it was essential to develop a vaccine that demonstrated efficient efficacy within a short period after vaccination. This decision is also supported by related studies on COVID-19 vaccine booster*

doses, where both BNT162b2²⁴ and ZF2001²⁵ showed favorable efficacy profiles 7 days post-vaccination."

References:

19. Baden LR, et al. Efficacy and Safety of the mRNA-1273 SARS-CoV-2 Vaccine. *N Engl J Med* **384**, 403-416 (2021).
20. Emary KRW, et al. Efficacy of ChAdOx1 nCoV-19 (AZD1222) vaccine against SARS-CoV-2 variant of concern 202012/01 (B.1.1.7): an exploratory analysis of a randomised controlled trial. *Lancet* **397**, 1351-1362 (2021).
21. Logunov DY, et al. Safety and efficacy of an rAd26 and rAd5 vector-based heterologous prime-boost COVID-19 vaccine: an interim analysis of a randomised controlled phase 3 trial in Russia. *Lancet* **397**, 671-681 (2021).
22. Voysey M, et al. Safety and efficacy of the ChAdOx1 nCoV-19 vaccine (AZD1222) against SARS-CoV-2: an interim analysis of four randomised controlled trials in Brazil, South Africa, and the UK. *Lancet* **397**, 99-111 (2021).
23. Wang XY, et al. Efficacy of heterologous boosting against SARS-CoV-2 using a recombinant interferon-armed fusion protein vaccine (V-01): a randomized, double-blind and placebo-controlled phase III trial. *Emerg Microbes Infect* **11**, 1910-1919 (2022).
24. Moreira ED, Jr., et al. Safety and Efficacy of a Third Dose of BNT162b2 Covid-19 Vaccine. *N Engl J Med* **386**, 1910-1921 (2022).
25. Dai L, et al. Efficacy and Safety of the RBD-Dimer-Based Covid-19 Vaccine ZF2001 in Adults. *N Engl J Med* **386**, 2097-2111 (2022).

Figures and Tables:

Figure 4: Solicited Adverse Events - would consider change the y axis scale. It's difficult to see the findings.

Response: As per your suggestion, Figure 4 was revised as follows:

A. Solicited Local Adverse Reactions

B. Solicited Systemic Adverse Reactions

Reviewer #2 (Remarks to the Author):

In this study entitled “Efficacy of the Tetravalent Protein COVID-19 Vaccine, SCTV01E: a Phase 3 Double-blind, Randomized, Placebo-controlled Trial,” the authors report the efficacy results of the Phase III COVID-19 vaccine trial.

In this study, the authors compare vaccine efficacy of SCTV01E, a tetravalent subunit vaccine containing trimeric spike protein for Alpha, Beta, Delta, and Omicron BA.1 variants adjuvanted with a squalene-based oil-in-water emulsion to placebo in an immune population (vaccinated or previously infected). These studies follow up the results of a phase III immunogenicity trial ([https://www.thelancet.com/journals/eclinm/article/PIIS2589-5370\(23\)00372-3/fulltext](https://www.thelancet.com/journals/eclinm/article/PIIS2589-5370(23)00372-3/fulltext)), which showed enhanced neutralization titers relative to inactivated vaccine or bivalent vaccine, and another study comparing SCTV01E to monovalent Pfizer vaccine showing improved Omicron vaccine-induced neutralization titers. The study is well performed and that the manuscript overall is well written.

The author have addressed most of my issues from previous iteration of this manuscript. The

major issue below that I have with the study cannot be addressed by the authors, however they have added a sentence to the discussion of why they used a saline control, and provided some al be it flawed) comparison to other studies.

Major Issues

Q1. While this study has already been completed and was reviewed by an IRB, I am not sure the ethics of a December 2022 placebo-controlled trial for a COVID-19 booster vaccine. By Dec 2022, it was clear that vaccines prevent infections and disease caused by SARS-COV-2. Whether ethical or not, by comparing to saline, rather than an inactivated vaccine or an mRNA vaccine, even without and updated variant, limit the utility of the results. The real-world decision is not whether to vaccinate, it is what to vaccinate with. Thus, the results of this study have limited impact, as there was no comparator vaccine.

Response: Thank you for your concerns regarding the ethics of the placebo-controlled trial for a COVID-19 booster vaccine conducted in December 2022. The decision for this design was made in alignment with WHO and Chinese guidelines for evaluating the protection of COVID-19 vaccines, considering the prevailing circumstances in China at the time. As of the late December 2022 cutoff date, no COVID-19 vaccines targeting SARS-CoV-2 Omicron variants had been approved in China, and mRNA COVID-19 vaccines were not authorized for use in the country. Both the WHO and authorities in China recognize the appropriateness of a placebo control group under certain circumstances, especially when there are sound reasons to doubt the efficacy of licensed vaccines.

We acknowledge that the absence of a comparator vaccine may limit the generalizability of the results to real-world scenarios. This limitation is addressed in the revised manuscript:

"Comparing our vaccine to saline rather than an inactivated or mRNA vaccine may limit the study's impact. The decision for a placebo-controlled design was influenced by the challenges of licensed COVID-19 vaccines in China, given the limited available evidence supporting their efficacy against the new variants during the study period. "

Minor issues

Q2. The authors should rewrite the sentence Lines 143-145:

Current phrasing

"All participants received COVID-19 vaccines based on the ancestral strain before enrollment. As required by the reviewers, the information regarding the prior vaccination is provided in Table S2."

Suggested New phrasing

"All participants received COVID-19 vaccines based on the ancestral strain before enrollment, and their prior vaccination history is provided in Table S2."

Response: As per your suggestion, we have revised the sentence as: *"All participants received COVID-19 vaccines based on the ancestral strain before enrollment, and their prior vaccination is provided in Table S3."*

Q3. The vaccination history in Table S2 should sub-grouped based on placebo or treatment

group

As per your suggestion, Table S3 has been revised as:

Table S3. Vaccine Sequences of Priming and Booster Vaccine

1 st dose	2 nd dose	3 rd dose	Number	Percentage (%)		
				Total	Placebo	SCTV01E
Inactivated vaccine			3	0.03%	0	0.07%
Adenovirus-vector vaccine			14	0.15%	0.20%	0.11%
Protein-based vaccine	Protein-based vaccine		2	0.02%	0	0.04%
Inactivated vaccine		Inactivated vaccine	1	0.01%	0.02%	0
Inactivated vaccine	Inactivated vaccine		429	4.67%	4.50%	4.83%
Adenovirus-vector vaccine	Inactivated vaccine		1	0.01%	0	0.02%
Adenovirus-vector vaccine	Adenovirus-vector vaccine		119	1.29%	1.28%	1.30%
Protein-based vaccine	Protein-based vaccine	Protein-based vaccine	1669	18.15%	18.19%	18.10%
Protein-based vaccine	Inactivated vaccine	Protein-based vaccine	1	0.01%	0	0.02%
Protein-based vaccine	Inactivated vaccine	Inactivated vaccine	3	0.03%	0.04%	0.02%
Inactivated vaccine	Inactivated vaccine	Protein-based vaccine	524	5.70%	5.53%	5.87%
Inactivated vaccine	Inactivated vaccine	Inactivated vaccine	6321	68.74%	69.03%	68.44%
Inactivated vaccine	Inactivated vaccine	Adenovirus-vector vaccine	108	1.17%	1.15%	1.20%
Adenovirus-vector vaccine	Adenovirus-vector vaccine	Adenovirus-vector vaccine	1	0.01%	0.02%	0

Q4. In the discussion the author reference the CDC report of vaccine effectiveness (ref 17) however the data is presented as a comparison to the current study. This should be changed as the 2 studies should not be compared head to head for several reasons, including 1) populations are quite different in terms of vaccine history in USA and China, 2) Assess vaccine efficacy at 2-3 months (ref 17) vs 7 or 14 days in the current study.

Response: Thank you for bringing this concern for us. Considering the differences in the

populations of the two studies in terms of vaccination history between the USA and China, and the impracticality of comparing vaccine efficacy head-to-head from two different studies, we have deleted this paragraph in the revised manuscript

Reviewer #4 (Remarks to the Author):

The authors have incorporated all the suggested corrections from the earlier version and I have no further comments.